# Explore Ultrasonic-Induced Mechanoluminescent Solutions towards Realising Remote Structural Health Monitoring

**DOI:** 10.3390/s24144595

**Published:** 2024-07-16

**Authors:** Marilyne Philibert, Kui Yao

**Affiliations:** Institute of Materials Research and Engineering (IMRE), Agency for Science, Technology and Research (A*STAR), 2 Fusionopolis Way, Singapore 138634, Singapore

**Keywords:** structural health monitoring, mechanoluminescence, ultrasonic guided wave, remote sensing, piezoelectric transducer

## Abstract

Ultrasonic guided waves, which are often generated and detected by piezoelectric transducers, are well established to monitor engineering structures. Wireless solutions are sought to eliminate cumbersome wire installation. This work proposes a method for remote ultrasonic-based structural health monitoring (SHM) using mechanoluminescence (ML). Propagating guided waves transmitted by a piezoelectric transducer attached to a structure induce elastic deformation that can be captured by elastico-ML. An ML coating composed of copper-doped zinc sulfide (ZnS:Cu) particles embedded in PVDF on a thin aluminium plate can be used to achieve the elastico-ML for the remote sensing of propagating guided waves. The simulation and experimental results indicated that a very high voltage would be required to reach the threshold pressure applied to the ML particles, which is about 1.5 MPa for ZnS particles. The high voltage was estimated to be 214 Vpp for surface waves and 750 Vpp for Lamb waves for the studied configuration. Several possible technical solutions are suggested for achieving ultrasonic-induced ML for future remote SHM systems.

## 1. Introduction

### 1.1. Structural Health Monitoring

Structures are subjected to diverse dynamic loads and environmental conditions. Maintenance is conventionally performed using Non-Destructive Testing (NDT) involving a skilled workforce conducting time-consuming and costly manual inspections prone to human errors. In the recent years, the process of structural maintenance and repair has shifted towards automation, driven by the escalating complexity and ageing of engineering structures and by the limited workforce capacity and high cost. As the demand for effective monitoring and early detection of structural damage increases, structural health monitoring (SHM) emerges as a promising solution for in-service health assessment. Such continuous monitoring offers real-time data acquisition, facilitating prompt decision-making. SHM solutions provide proactive effective maintenance strategies, enabling targeted repairs and interventions that maximise structures’ lifespan while minimising downtime. Therefore, the long-term objective of the SHM solution is to improve safety and longevity, and to reduce maintenance costs and shut-down time, despite the cost of installation of a highly reliable SHM system with advanced sensors. Moreover, the incorporation of smart structures introduces adaptive and resilient solutions. SHM systems are not only in use in civil infrastructures [1], for bridges and buildings, but are also developed for applications in other industries such as aerospace and mechanical engineering structures, such as aircraft or pipes.

Some of the challenges for implementing SHM solutions are the wire connections, weight of the sensors and wires, installation time, and alignment errors. The need for remote or wireless solutions in SHM arises from expensive and complex installations of many sensors over large areas and from accessibility issues. With many critical structures located in remote or inaccessible areas, using remote SHM solutions by integrating wireless sensors allows for reduced on-site inspections associated with a high cost. Wireless sensor networks enable a reduced installation time and simplified maintenance and facilitate the deployment of many sensors, empowering SHM systems with more scalable and more flexible solutions with access to data from various locations [2]. The development of wireless SHM solutions can be facilitated by integrating Internet of Things (IoT) technologies for continuous connectivity and real-time data collection and transmission or using edge computing technologies for efficient data processing closer to the source, enabling reduced data transmission [1]. Ferroelectric materials and devices are highly suitable for applications in wireless SHM due to their ability to convert stimuli to signals without an external power supply, making them energy-autonomous sensors [3]. They can be used to produce ultralow power and even self-powered wireless smart sensors for detecting and measuring a wide variety of parameters, such as temperature, pressure, and vibration.

Ultrasonic-based SHM techniques, developed as an extension of ultrasonic NDT, provide high sensitivity to defects or any irregularities with rapid response times. Guided waves are used in ultrasonic SHM applications because they can travel long distances with minimal attenuation and can propagate into complex geometries. They are usually generated and detected by ultrasonic piezoelectric transducers. A well-known technique for the remote sensing of guided waves is to use a laser vibrometer, which can accurately measure the out-of-plane surface velocities during guided wave propagation [4]. This contactless inspection method could be implemented as scheduled SHM maintenance, allowing for a minimised number of fixed transducers to be installed on the structure for guided wave generation only [5]. Moreover, a laser scanning vibrometer enables the straight-forward interpretation of guided wave propagation for damage localisation and characterisation.

### 1.2. State of the Art of Mechanoluminescence-Based SHM

Mechanoluminescence (ML) is a phenomenon where light emission is produced when a material undergoes mechanical stress. Because the energy of emitted photons during solid deformation is more than two orders of magnitude higher than the average energy supplied to an atom, ion, or molecule, it is assumed that the energy supplied to an atom, ion, or molecule during mechanical deformation is not uniformly distributed [6]. This is due to localised physical processes increasing the energy in specific regions of the solid, which can be classified according to the phenomenon involved, either deformation-induced (fracture, plastic, or elastic deformation of solids) or tribo-induced due to the contact phenomenon (electrically, chemically, or thermally). The different ways to induce ML can be found in the literature [6]. Elastico-ML offers the advantage of linear, reusable, and non-destructive light emission, which is essential for sensor applications [7]. The most studied ML materials for their high light intensity are phosphors such as SrAl_2_O_4_:Eu^2+^ (SAOE) or SrAl_2_O_4_:Eu^2+^, Dy^3+^ (SAOED) with green light emission, manganese-doped zinc-sulfide (ZnS:Mn^2+^) with orange light emission, and copper-doped zinc sulfide (ZnS:Cu) with blue or green light emission [7,8,9]. In contrast to SAOE, metal ion-doped ZnS provides a self-reproducible ML effect, meaning that the light intensity is maintained under repetitive mechanical stimuli without the need for light pre-irradiation [10]. Apart from the reproducibility, the good linearity due to the strong correlation of the ML intensity with applied stress [8,10] is desired for sensing applications.

Some ML technologies applied for SHM or NDT in the literature have been reviewed, and it was found that the main application of ML-based SHM was for strain monitoring. Stress sensing applications were previously reviewed [8]. In a recent study, the progressive failure of the epoxy adhesive layer between carbon fibre-reinforced plastics (CFRPs) and aluminium was visualised using ML paint made of SAOE and epoxy sprayed on the CFRP [11]. Light emissions were recorded by CCD (charge-coupled device) cameras while a cross tension was applied to the CFRP and aluminium samples pulled in opposite directions. By observing the different stress concentration behaviours, crack growth, and fracture, the test design could be optimised. Similarly, an ML coating made with SAOE in epoxy and a digital image correlation (DIC) method was used to effectively measure the strain field within an aluminium sample under tension testing due to the linear relationship between the effective strain and ML intensity [12]. In another work, sensors were developed and embedded into glass fibre-reinforced plastics (GFRPs) or CFRP composite structures for barely visible damage (BVID) detection and localisation with great potential for SHM [13]. The pressure sensor is composed of an ML layer, which is made with ZnS:Cu ML crystals embedded in polydimethylsiloxane (PDMS), for light emission due to the impact and a perovskite layer for converting light photons into an electric current.

There are some applications of the SHM of large-scale infrastructures using an ML paint made with SAOE in epoxy reported in the literature. A 10 min historical-log recording system was implemented on a 50-year-old bridge in use [14]. The paint was applied on a visible crack and a photosensitive film was put on top to obtain a 2D stress distribution map representing the crack growth. The crack growth on the bridge was effectively monitored in this method without requiring electricity. In another application, the paint was applied on the intersection parts of different U-ribs of a highway bridge and a CCD camera was used to capture the emitted light [15]. This method could quantitatively evaluate and image dynamic stress for fatigue cracks and stress concentration detection.

Ultrasound-induced ML applications were found in the literature. Combined with targeted focused ultrasound (FUS) excitation, ML nanoparticles injected to specific tissues or organs or inhaled can be used for potential non-invasive bioimaging applications in biomedical research [16]. Other applications were reported for ML film in regular ultrasonic water bath cleaner found in the lab. Films made of (Ca,Sr)ZnOS:Mn^2+^ phosphors [17], porous films made with ZnS:Cu,Mn in poly(vinylidene fluoride-co-hexafluoropropylene) (PVDF-HFP) [18], and films made of SAEO [19] showed the intensity distribution of ultrasonic waves inside an ultrasonic cleaner at about 40 kHz. Finally, ultrasound-induced ML was observed on piezoelectric transducers. A scanning gel was used to ultrasonically couple a 100 μm thick ML film made of SAOE in epoxy and a piezoelectric transducer operating at 20 MHz [20]. The ultrasonic output power distribution on the surface of the transducer was successfully visualised and quantified by recording images with a CCD camera. A piezo-phototronic luminescence device was developed by using a Pb(Mg_1/3_Nb_2/3_)O_3−x_PbTiO_3_ (PMN-PT) single-crystal substrate with (001) orientation and an ML film made with ZnS:Al,Cu,Mn in PDMS [21]. However, a high voltage of 400 Vpp and frequencies lower than 1 kHz were used for demonstration, and ultrasonic frequencies are yet to be tested. The 3D reconstruction of an ultrasound pressure field of an ultrasonic transducer operating at 3.3 MHz was obtained using an ML membrane made with BaSi_2_O_2_N_2_:Eu able to acoustically produce piezo-luminescence [22,23]. The method was fast, easy, and without need for expensive equipment.

### 1.3. Potential of Ultrasonic-Induced ML for Remote SHM

Our literature review has shown that ML-based SHM applications were performed for structures under loading due to the light emission being a dynamic phenomenon sensitive to change in the strain. The light emission also requires a certain threshold level of stress. In general, SHM applications showed crack monitoring performed through passive sensing. However, passive approaches are exposed to various noise and require continuous data recording. In contrast, ultrasonic active sensing approaches are fully controlled as a known signal would be transmitted to directly interrogate the structural state at a known time. This assists signal interpretation and permits discontinuous regular data recording. Furthermore, ML-based ultrasonic applications in the literature were performed in vivo using FUS, or in water using bulk waves, or directly on top of a piezoelectric transducer. There is no application of ML techniques for detecting ultrasonic guided wave propagation along a structure for active SHM applications to our best knowledge. 

This study explores for the first time the feasibility of utilising ultrasonic-induced ML technologies for self-powered remote SHM. A composite coating made of ML particles and resin can be dispersed onto the structure, wherein each ML particle could act as a mechanical sensor emitting light when the guided wave propagates, as schematically illustrated in Figure 1. The proposed method could help understand and interpret a wave interaction with a defect. One of the main requirements for SHM applications is to minimise intrusion by using lightweight technologies without a hazard. The benefits of remote SHM using ML solutions are the potential ability to visualise ultrasonic guided wave propagation from spatially distributed light emissions for user-friendly, fast, and easy interpretation and the non-contact feature without requiring cabling with the reduced weight and installation complexity. However, the proposed method still requires installing ML material in the form of coating and line-of-sight access for detecting the light emitted (human eye, camera, or another optical detector).

In this work, the feasibility of our proposed methodology of ultrasonic-based ML for remote SHM is first investigated through a mechanism study on mechanoluminescent materials. Then, the results of the guided wave visualisation based on ML are presented, analysed, and discussed. Finally, other ML-based SHM techniques for active sensing are suggested and compared with the presented technology.

## 2. Theory

### 2.1. Mechanisms behind Elastico-ML and Material Selection

The different possible mechanisms and theory behind the ML of different materials from plastic and elastic deformation have been explained in the literature [24,25]. The mechanism for ML is based on the de-trapping of carriers in phosphors by lowering the energy barriers under dynamic loading, which is caused either by reducing the band gap of the host or by changing the positions of the defects in the band gap of the host. Besides the trap-controlled mechanism under plastic deformation, another mechanism has been explored for elastico-ML. Because a dramatic change in the band gap of the host is not likely under a low pressure below around 10 MPa, elastico-ML was explained by the de-trapping of electrons in the presence of a local piezoelectric field [25,26]. This so-called piezo-photonic effect allows for the self-recovery elastico-ML of doped-ZnS, such as ZnS:Cu and ZnS:Mn. The local piezoelectric field is produced under ZnS crystal deformation due to the non-centrosymmetric crystal structure of ZnS. Therefore, it was found that using ML coating made of ZnS:Cu,Mn embedded in PVDF could enhance the ML intensity due to the contribution of an external piezoelectric field provided by PVDF under pressure, which can reduce the trap depth or tilt the band structure of ZnS [18,27,28]. Moreover, it was found that the ML intensity could be enhanced by enhancing the mechanisms of stress transfer from the hydrophobic PDMS matrix to rigid ZnS:Cu ML particles using a moisture-resistant aluminium oxyhydroxide Al_2_O_3_ coating on the surface of ZnS:Cu particles [29,30], which could be applied to a PVDF-ZnS composite coating as well.

The guided wave propagation would induce elastic deformation; therefore, elastico-ML could be enhanced by using a PVDF matrix, which can enhance the piezo-photonic effect. Al_2_O_3_-coated ZnS:Cu particles embedded in PVDF were selected as the ML coating because of the possible enhanced stress transfer and because of the high-intensity, self-reproducible, and repeatable light emission of robust ZnS. A higher concentration of ZnS:Cu particles in the ML coating is desired because it would imply a higher ML intensity. However, an excessively high concentration would also reduce adhesion to the substrate and increase opacity, which could block the light emission. The resulting ML coating with a concentration of about 25 vol% showed acceptable adhesion to the aluminium plate.

### 2.2. Threshold Pressure

ML is controlled by the pressure applied to the ML particles. Because guided wave propagation induces elastic deformation, we have analysed the threshold pressure magnitude for achieving elastico-ML. Some threshold pressure values found in the literature are summarised in Table 1. The lowest pressure values for the elastico-ML appearance were found to be about 230 kPa [26]. However, this value represents the lower bound of a distribution of threshold pressures experimentally observed across forty-four different single ZnS:Mn microparticles. The test showed that some particles were more sensitive than others, with only 12% of the ZnS:Mn microparticles tested emitting light at a pressure below 1 MPa. From these results, it can be understood that the threshold pressure should be observed as a range of pressure rather than a hard threshold. For the ML coating made from ZnS:Cu embedded in PVDF, it can be assumed that the threshold pressure for elastico-ML due to guided wave propagation would be about 1.5 MPa [31].

## 3. Results

### 3.1. Modelling and Simulation Results

A 2D model was made with COMSOL Multiphysics (version 6.1) representing a 1.6 mm thick aluminium alloy structure to monitor, the piezoelectric ceramic transducer made of lead zirconate titanate (PZT) with an effective piezoelectric d_33_ coefficient of 400 pm/V, and the ML coating, as shown in Figure 2a. The coating was made of polyvinylidene fluoride (PVDF) and a row of round ZnS particles of diameter 28 μm regularly spaced along the coating. The spacing of the particles represents a specific concentration of ZnS particles in the ML coating. Three different concentrations in volume of ZnS particles in ML film were tested, i.e., 18 vol%, 24 vol%, and 32 vol%. The PZT transducer was transmitting a continuous sine wave at a specific frequency. Frequencies from 1 kHz to 10 MHz were tested. The model simulation was developed to study the effect of different parameters, i.e., the concentration of ZnS particles in the ML film and the frequency and the voltage of the ultrasonic signal generated by the PZT transducer, to optimise the pressure applied to the ML ZnS particles. The displacement on the top surface of the ML film and the pressure applied to ZnS particles at different distances from the PZT transmitter were obtained, as illustrated in Figure 2a. As expected, the displacement and pressure values were proportional to the voltage applied. Hence, the values reported are based on the voltage of the excitation signal, and the results of the maximum displacement and maximum pressure according to the frequency of the transmitted sine wave are shown in Figure 2b and Figure 2c, respectively.

The ZnS concentration of the ML film had little effect on the displacement values. For frequencies below 3 MHz, a higher ZnS concentration allowed for higher pressure to be applied. In general, and despite resonance around 6 kHz and 100 kHz, lower frequencies induced higher displacement, while higher frequencies induced higher pressure applied on the ZnS particles. This would suggest surface wave propagation promotes higher pressure on the top surface of the plate, which could be expected as the energy of the surface waves are mainly concentrated around the surface of the structure. However, the higher pressure at higher frequency was found to be localised near the transmitter, as observed in Figure 3b for a concentration of ZnS of 24 vol%. Similar trends were observed for concentrations of 18 vol% and 32 vol%.

### 3.2. Experimental Results

A sample was made with an ML coating on top of a 1.6 mm thick aluminium plate and with a PZT ceramic transducer, P-876.SP1 DuraAct (PI Ceramic GmbH, Lederhose, Germany), bonded on the plate. The ML solution was prepared with 1 g ZnS:Cu powder with an average diameter of 28 μm in 4 mL of solution of 12 wt% P(VDF-TrFE) (70/30) in dimethylformamide (DMF) and Acetone (2:8). The ML solution was coated on the plate and annealed at 135 °C on a hot plate. The resulting ML coating had a concentration of about 25 vol% and showed acceptable adhesion to the plate. For validating the simulation model, guided wave-induced displacements were compared with the experimental results. The PZT transducer was transmitting a 20 Vpp continuous sine wave at different frequencies, and the displacements were measured using a laser vibrometer (PSV-400 from Polytec South-East Asia Pte Ltd., Singapore). An area of the top surface of the ML coating was scanned, and the average displacement over a 10 mm width of the coating was calculated for every 1 mm over the length of the coating. The resulting displacement values per voltage of the excitation signal for both the experimental and simulation results were then plotted according to the distance from the transmitter for the excitation signal at 40 kHz, 100 kHz, and 250 kHz, as presented in Figure 4. While the displacements along the propagation direction of the guided wave generated by the transmitter showed a different pattern and values, the order of magnitudes were similar enough to validate the simulation results. Figure 2b shows some vibration phenomenon in simulation results at 100 kHz that were not observed in experimental results. This can be due to uncertain mechanical clamping or coupling conditions or other assumptions in the simulation, such as physical contacting interfaces. From the simulation results at 40 kHz and 250 kHz, the displacements are different, but the pressure values applied to the ML particles are similar. The pressure applied to the ML particles during experiments could probably be estimated to be about 0.8 kPa/Vpp.

The ML coating made of P(VDF-TrFE) was poled by a non-contact corona discharge gun at 17 kV. Light emission was observed during this high-voltage poling process in multiple forms. First, turning on or off the corona gun induced a brief but bright green light emission over a large surface of the ML coating, as observed in Figure 5a. Moreover, a blue light emission was also observed at some localised points over the surface of the ML coating, shown in Figure 5b, and a green light emission could be observed at a certain distance from the tip of the corona discharge. This green light was radiantly emitted and was disappearing when the tip of the corona discharge was further away as observed in Figure 5c.

Ultrasonic testing was conducted using a PZT transmitter bounded on the aluminium plate with an ML coating on its surface to generate guided waves. A continuous sine wave was generated at different frequencies with a voltage of 100 Vpp. However, no light was observed from the guided wave propagation. Only a blue light emission was observed at some localised points, as observed during the corona poling. These blue dots were observed at different positions over the surface of the ML coating depending on the frequency generated. These observations are further discussed in the following section.

## 4. Discussion

The experimental results indicated that the propagation of the guided wave generated by the PZT transmitter within the aluminium plate could not initiate enough pressure to achieve elastico-ML from the ZnS:Cu particles. Practically, it was not feasible to apply a voltage higher than 100 Vpp to the PZT due to the vibration damaging the electrical connections. From the threshold pressure values in Table 1, a minimum pressure applied to the ZnS:Cu particles of 1.5 MPa is required to achieve elastico-ML [31], corresponding to 15 kPa/Vpp at 100 Vpp. From the simulation results in Figure 2c, such pressure could not be reached. Potentially, using higher voltage or a different piezoelectric transmitter with a higher piezoelectric coefficient, elastico-ML may be achieved with surface waves above 3 MHz, as the energy of surface waves is mainly on the surface of the structure where the ML coating is. However, this would require good adhesion of the film to the plate to allow for the transfer of energy. Indeed, the perfect bounding of materials to each other, i.e., between ZnS and PVDF, between PVDF and aluminium, and between PZT and aluminium, was assumed in the simulation model, implying the best mechanical transfer possible. Furthermore, it was observed experimentally and shown in Figure 5 that applying a high voltage from the corona discharge to the ML coating enabled light emission.

Moreover, ZnS:Mn could be used instead of ZnS:Cu as ML particles for its higher sensitivity, attributed to its lower threshold pressure. However, it was found that the threshold pressure may differ for each ML particle [26]. The threshold pressure is the minimum pressure to allow for ML mechanisms to happen. Thus, the threshold pressure must be reached. Once it is reached, if the elastico-ML intensity is low and the light emission is not visible to the human eye, an optical detector may be considered in the future. Optical fibres may also be considered to transfer light from the sensing location in case of accessibility issues.

Due to the low piezoelectric coefficient of PVDF, it can also be assumed that the piezoelectric field from PVDF was not strong enough to enhance the piezo-photonic effect, as found in other studies [18,27,28]. By embedding ceramic particles within the ML coating to enhance the piezoelectric field surrounding the ML particles, it can be thought that the piezo-photonic effect may be enhanced as well. However, ceramic particles have a high density, suggesting a higher weight for the ML coating, which is not desired for SHM applications.

The experimental observations during the corona poling in Figure 5b and during ultrasonic testing with PZT transmitting guided waves revealed blue light emissions in the form of localised dots. This is due to electroluminescence, which is an electrically induced light emission. Indeed, ZnS:Cu emits green light from ML, while it emits blue light from electroluminescence [35]. By embedding ZnS:Cu electroluminescent material in a piezoelectric coating such as PVDF, the direct piezoelectric effect could be used, wherein the mechanical energy from the guided wave propagation is converted by PVDF into electrical energy, thus inducing electroluminescence. However, the piezoelectric coefficient of PVDF may be too low to capture this fully. A transparent top electrode could be deposited on top of the ML coating to further study the potential of using electroluminescence for guided wave propagation visualisation.

The advantages of using ultrasonic-induced elastico-ML over conventional sensors for SHM include self-powered sensing, remote and wireless sensing, the flexibility and adjustability of surface coating, spatially distributed sensing, and the real-time visualisation of a stress distribution or of ultrasonic wave propagation. The visual observation of guided wave propagation is of great advantage for straight-forward defect detection and interpretation of a guided wave interaction with a defect. The main competing technology to the proposed ML technology would involve using a laser vibrometer. While a laser vibrometer is a well-known effective technology for guided wave propagation visualisation without the need for preparing the surface of the structure to monitor [36,37], it is highly expensive and bulky.

This concept was applicable in principle for the SHM of large structures, hence the use of guided waves able to travel long distances. However, the current experimental results showed insufficient pressure to achieve visible ML even at a short distance. Therefore, it is not practical yet to realise ultrasonic-induced mechanoluminescent remote SHM over a large area without further substantially improving the mechanoluminescent efficiency. Nevertheless, this could be used for local hotspot monitoring.

The PVDF coating used in our experiments previously showed the robustness and resilience in harsh environmental conditions, including high and low temperature cycling, icing, rain, humidity, and the salt fog test, when using a protective layer [38]. Non-contact corona poling could be used for polarising the PVDF coating, while it is probable the poling process was not a necessary step for achieving ML.

## 5. Conclusions

Herein, we proposed a new method for ultrasonic-based SHM applications using ML and guided waves. From our review of the relevant applications of ML-based monitoring, passive methods using the dynamic monitoring of structures under load-induced stress and ultrasonic applications using bulk waves directly next to the piezoelectric transmitter were found. However, we did not find any reports in the literature concerning active ultrasonic guided wave sensing using ML for remote SHM applications. We proposed the use of ML particles such as ZnS:Cu embedded into a piezoelectric coating made of PVDF and coated onto the surface of the structure. By generating guided waves using an ultrasonic transmitter, such as a PZT transducer, the propagation of the guided wave will induce elastic deformation to achieve elastico-ML by the ML particles. The main mechanism explaining elastico-ML is the piezo-photonic effect that can be enhanced using a piezoelectric matrix, such as PVDF. The feasibility of using ML particles for active ultrasonic-based SHM was evaluated through preliminary experiments and a theoretical simulation. The feasibility for guided wave visualisation based on ML was discussed. From the experimental and simulation results, we deducted that the inability to visualise the guided wave propagation in an aluminium structure using ZnS:Cu embedded in PVDF is due to the low pressure or low piezoelectric field applied by PVDF to the ML particles. In the current experimental configuration, the highest pressure was about 7 kPa/Vpp at 7 MHz for surface waves and about 2 kPa/Vpp at 100 kHz for Lamb waves. With the threshold pressure being about 1.5 MPa for ZnS particles, a very high voltage of at least 214 Vpp at 7 MHz or 750 Vpp at 100 kHz would be required. Further work could be performed for enhancing the performances of the ML film for achieving the visualisation of guided waves propagating in a structure. Possible solutions may include using more sensitive ML particles such as ZnS:Mn or SAOE; enhancing the pressure applied to the particles using a higher voltage or a bulkier piezoelectric transmitter with a higher piezoelectric coefficient; and enhancing the piezo-photonic effect using embedded ceramic piezoelectric particles. Our proposal and investigation outcome in this work present the technical challenges and show the direction for the future efforts towards ultrasonic-based remote SHM applications enabled by the combination of ML and guided waves.

## Figures and Tables

**Figure 1 sensors-24-04595-f001:**
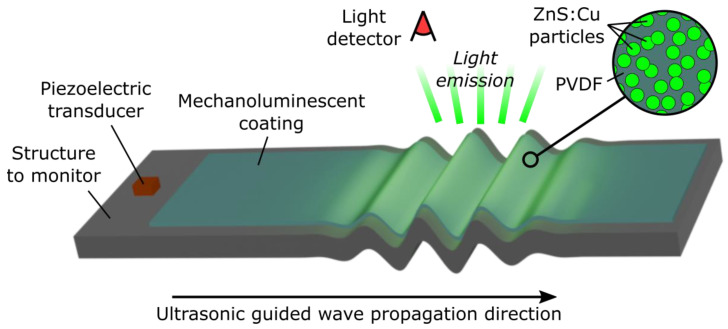
The schematic illustration of the proposed ML-based remote ultrasonic SHM solution.

**Figure 2 sensors-24-04595-f002:**
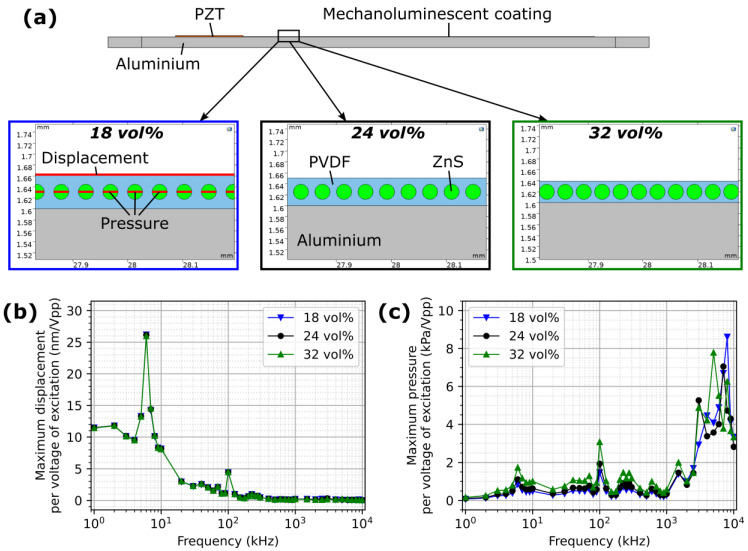
(**a**) A schematic illustration of the 2D model for three different concentrations in volume of ZnS particles in PVDF in the composite ML film, the corresponding results per voltage of the excitation signal showing (**b**) the maximum displacement obtained on top of the ML coating, and (**c**) the maximum pressure obtained on the ZnS particles, according to the frequency of the excitation signal.

**Figure 3 sensors-24-04595-f003:**
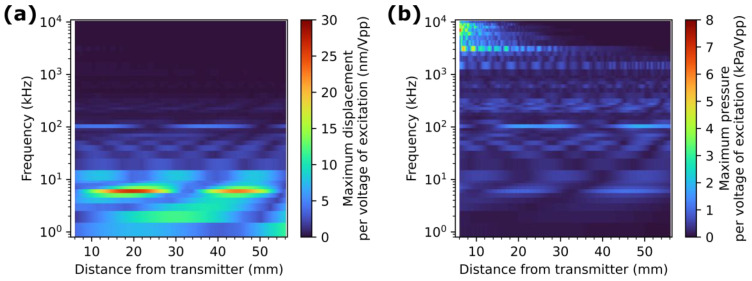
The simulation results per voltage of the excitation signal for the 24 vol% concentration of ZnS results showing (**a**) the maximum displacement obtained on top of the ML coating and (**b**) the maximum pressure obtained on the ZnS particles, according to the frequency of the excitation signal and to the distance from the transmitter.

**Figure 4 sensors-24-04595-f004:**
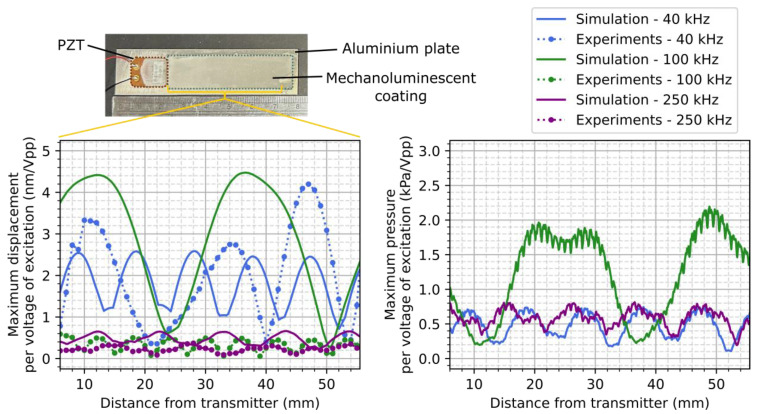
A comparison of the experimental and simulation displacements obtained on top of the ML coating, and the simulation pressure applied to the ZnS particles, along the propagation direction per voltage of the excitation signal for a 24 vol% concentration of ZnS, at an excitation frequency of 40 kHz, 100 kHz, and 250 kHz.

**Figure 5 sensors-24-04595-f005:**
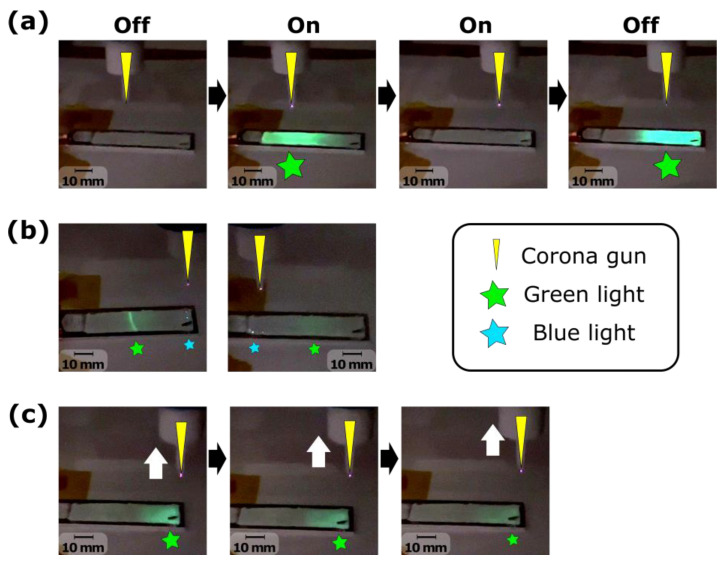
Photos of the observations during the corona poling with (**a**) the effect of turning on or off the corona discharge showing an intense flash of green light, (**b**) different light emission during the poling with the light emission from a certain horizontal distance from the corona discharge, and (**c**) the effect of the vertical distance from the corona discharge showing a radiant light emission with white arrow showing the movement of the corona gun.

**Table 1 sensors-24-04595-t001:** Threshold pressure for achieving elastico-ML for different materials.

ML Material	Threshold Pressure for Elastico-ML (MPa)	Reference
SAOE	1	[9]
SAOED (ceramic coating)	0.45	[32]
ZnS:Cu,Cl microparticles	1.5	[31]
ZnS:Mn microparticles	0.8	[31]
	0.6 to 0.65	[33,34]
	0.23 to 47.15	[26]

## Data Availability

The data presented in this study are available on request from the corresponding author.

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
