# Peer review of "Explore Ultrasonic-Induced Mechanoluminescent Solutions towards Realising Remote Structural Health Monitoring"

_sensors, 2024, doi:10.3390/s24144595_

Round 1

Reviewer 1 Report

Comments and Suggestions for Authors

In this paper, this work proposes a method for remote ultrasonic-based structural health monitoring using mechanoluminescence, through simulation experiments and compared with field experiments.provides the possibility of realizing remote SHM in the future The questions are as follows:

(1)  In page 5 line 204-210, only 12% of the ZnS:Mn microparticles tested emitting light at a pressure below 1 MPa, however it assume that the threshold pressure for elastico-ML due to guided wave propagation would be about 1.5 MPa,will this result in errors?

(2)  In page 5 line204-210,Three different concentrations in volume of ZnS particles in ML film were tested, i.e., 18 vol%, 24 vol%, and 32 vol%,why not choose 18 vol%, 24 vol%, and 30 vol%?

(3)  In page 6 figure 2(b),why dose the maximum displacement obtained on top of the ML coating suddenly rise and fall?

(4)  In page 6 figure 2(c),multiple turning points appear in the figure,Contrary to the description in the text that "the higher frequencies induced higher pressure applied on the ZnS particles".

(5)  In page 7,suggest adding simulation results of ZnS with concentrations of 18vol% and 32vol%.

(6)  In page 7 line 270-273, the pressure values applied to the ML particle  is different from the other two when excitation signal at100 kHz,will choosing a pressure of 0.8 kPa/Vpp for experiments have an impact?

Author Response

Comment #1: In this paper, this work proposes a method for remote ultrasonic-based structural health monitoring using mechanoluminescence, through simulation experiments and compared with field experiments. provides the possibility of realizing remote SHM in the future The questions are as follows:

In page 5 line 204-210, only 12% of the ZnS:Mn microparticles tested emitting light at a pressure below 1 MPa, however it assume that the threshold pressure for elastico-ML due to guided wave propagation would be about 1.5 MPa, will this result in errors?

Our Reply:

Thank you for your interest and comments. These statements on threshold pressures are from different articles and for different particles, ZnS:Mn and ZnS:Cu. We are interested in the threshold pressure for realising elastico-mechanoluminescence for ZnS:Cu, which should be about 1.5 MPa. However, from the article on ZnS:Mn, we understand that the threshold pressure should be observed as a range of pressure rather than a hard threshold, as it was found that only a certain percentage of particles would emit light at a certain pressure. Therefore, yes, there could be errors on the threshold pressure. A sentence has been added in the manuscript page 5, line 214-215, to clarify this.

Comment #2: In page 5 line204-210, three different concentrations in volume of ZnS particles in ML film were tested, i.e., 18 vol%, 24 vol%, and 32 vol%, why not choose 18 vol%, 24 vol%, and 30 vol%?

Our Reply:

The three concentrations were chosen from the spacing of the ZnS particles, . This spacing was a certain percentage, , of the ZnS particles diameter, , i.e., , with , , and . The concentrations in volumes were then calculated to be respectively 32 vol%, 24 vol%, and 18 vol%.

Comment #3: In page 6 figure 2(b), why dose the maximum displacement obtained on top of the ML coating suddenly rise and fall?

Our Reply:

The maximum displacement peaked at about 6 kHz, and this can be interpreted as a natural frequency vibration of the structure, at which the structure vibrates intensely.

Comment #4: In page 6 figure 2(c), multiple turning points appear in the figure, Contrary to the description in the text that "the higher frequencies induced higher pressure applied on the ZnS particles".

Our Reply:

There are indeed some turning points around 6 kHz and 100 kHz, which could be impacted by resonance around these frequencies, however the pressure was still lower than 4 kPa/Vpp. Whereas the general trend after deducting the impacts of resonance is higher pressure at higher frequency, and the pressure reached values higher than 4 kPa/Vpp at frequencies higher than 3 MHz. The sentence was changed to highlight the resonance phenomenon line 248.

Comment #5: In page 7, suggest adding simulation results of ZnS with concentrations of 18vol% and 32vol%.

Our Reply:

We did not include simulation results for other concentrations as in Figure 3 because results from Figure 2 suggested little variations among concentrations of 18vol%, 24vol%, and 32vol%. Nevertheless, simulation results were observed, and similar trends were found for the different concentrations, as noted in the manuscript: “Similar trends were observed for concentrations of 18 vol% and 32 vol%.”

Comment #6: In page 7 line 270-273, the pressure values applied to the ML particle is different from the other two when excitation signal at 100 kHz, will choosing a pressure of 0.8 kPa/Vpp for experiments have an impact?

Our Reply:

The experimental results were done to confirm the order of magnitudes of the displacements measured. Therefore, we can confirm the order of magnitude of the pressure around 1 kPa/Vpp. It is not practical for this study to obtain exact values for the pressure, but this study was looking into order of magnitude to conclude on the feasibility in using ZnS particles for generating guided waves-induced light emission. As discussed in the first comment, the threshold pressure can be seen as a range of pressure and not an exact value. We hope this clarify the reason of choosing a pressure of 0.8 kPa/Vpp.

Reviewer 2 Report

Comments and Suggestions for Authors

This work is devoted to the remote structural health monitoring, which is a well-known and practically important scientific and engineering problem. Such problems are currently being successfully solved using commercially available sound emitters and sound sensors for monitoring the condition of buildings, bridges, pipelines and other objects. Section 1.2 reviews the State of the art of mechanoluminescence-based SHM. As shown in this section “In general, SHM applications showed crack monitoring performed through passive sensing” (page 4, line 140). However, this work is the first time to propose the use of Ultrasonic-induced mechanoluminescent solutions for this purpose. Unfortunately, from my point of view, this is impossible to do for a reason also described in the article itself. To induce the elastic-ML effect, it is necessary to create a sufficiently large “Threshold pressure”, as shown in Table 1. To create such a large mechanical stress, it is necessary to use a PZT transducer with an electrical voltage of hundreds of volts at a distance of only a few centimeters from the sound source to the light-emitting ML film! As the authors of the work themselves note, in the experiment no light was observed from guided wave propagation (page 9, line 296). It should also be taken into account that structures that are really interesting for such a technology can have dimensions of tens and hundreds of meters, and the attenuation of ultrasound in typical materials (steel, concrete) can reach several dB/m. This raises the requirement for the power of the ultrasonic emitter by tens of times. Unfortunately, the paper does not propose any method to overcome the above limitations. In fact, there is not a single formula in the text of the article at all, despite the finite element study carried out. Therefore, despite the great work done, this article cannot be recommended for publication, because the effect stated in the title was not realized.

Author Response

Comment #1: This work is devoted to the remote structural health monitoring, which is a well-known and practically important scientific and engineering problem. Such problems are currently being successfully solved using commercially available sound emitters and sound sensors for monitoring the condition of buildings, bridges, pipelines and other objects. Section 1.2 reviews the State of the art of mechanoluminescence-based SHM. As shown in this section “In general, SHM applications showed crack monitoring performed through passive sensing” (page 4, line 140). However, this work is the first time to propose the use of Ultrasonic-induced mechanoluminescent solutions for this purpose. Unfortunately, from my point of view, this is impossible to do for a reason also described in the article itself. To induce the elastic-ML effect, it is necessary to create a sufficiently large “Threshold pressure”, as shown in Table 1. To create such a large mechanical stress, it is necessary to use a PZT transducer with an electrical voltage of hundreds of volts at a distance of only a few centimeters from the sound source to the light-emitting ML film! As the authors of the work themselves note, in the experiment no light was observed from guided wave propagation (page 9, line 296). It should also be taken into account that structures that are really interesting for such a technology can have dimensions of tens and hundreds of meters, and the attenuation of ultrasound in typical materials (steel, concrete) can reach several dB/m. This raises the requirement for the power of the ultrasonic emitter by tens of times. Unfortunately, the paper does not propose any method to overcome the above limitations. In fact, there is not a single formula in the text of the article at all, despite the finite element study carried out. Therefore, despite the great work done, this article cannot be recommended for publication, because the effect stated in the title was not realized.

Our Reply:

We agree this work suggests that it is not practical yet to realize ultrasonic-induced mechanoluminescent remote structural health monitoring over a large area without further substantially improving the mechanoluminescent efficiency. We added sentences at the end of the discussion and at the end of the conclusion to reaffirm the little likelihood in using this technology for SHM of large structures. However, the potential for monitoring hotspots or some critical points does exist. In reference to the reviewer’ comments, we agree the title should be changed to: “Explore ultrasonic-induced mechanoluminescent solutions towards realizing remote structural health monitoring.”

Reviewer 3 Report

Comments and Suggestions for Authors

Authors propose a very interesting solution for SHM (Structural Health Monitoring) of structures. They suggest using mechanoluminescent materials to visualize the path of waves on the surface of a structure and use this information for SHM purposes (e.g., cracks detection or monitoring).

The paper represents a feasibility study of the concept, including a first attempt at experimental validation. This attempt failed, but the authors honestly present these results and seek to understand the reasons behind the failure of their initial trial. This paper and its content could therefore be very useful for those wishing to work in this direction and continue developing this solution.

I therefore recommend the publication of this paper, after some minor improvements.

Changes/improvements to be made before publication:

L73-81: explanations are not clear. Please improve.

L91: for "stress" or "strain" monitoring?

Some comments about wear? How would such a coating (PVDF + ML particles) resist harsch environmental conditions (e.g., heavy rain, hail, salt and small stones projections...)

How would PVDF be poled on a bridge? How long would the polarization "survive", in outdoor conditions?

Fig. 4 actually shows huge discrepancies between simulated and observed results. However, authors have the merit and honesty "to show it as it is".

323-325: you suggest using optical fibers to pick up and transfer the dim light emitted by the coatings... All in all, this complexifies the solution too much, for practical SHM applications.

Also... if you need to mount a highly powerful PZT transducer just next to the PVDF-ML coating to excite it, you would be better off installing a standard strain sensor directly where you want to measure the stress or strain state (e.g., vibrating wire sensor, DMS gauges etc...). It hard to see what your solution would bring in that case. But again, maybe there is a niche somewhere, for your solution.

So, please add, in the discussion section, a few sentences about what could realistically be done with this solution in the field of SHM. Coating a structure with PVDF-ML material seems highly unrealistic for a bridge (how would it be applied? Rapid wear issue...). Adding optical fibers to transfer a very weak light signal on a bridge is also impractical (complexity, cost, fragility of the system... competing with directly integrated optical fibers?). However, you may be able to identify some niche applications (buildings? tunnels? machines?...). That would be very interesting to know.

Comments on the Quality of English Language

Some sentences are a bit difficult to read. I recommend, if possible, having the text improved by a native speaker. Alternatively, conduct a detailed proofreading and improve sentences here and there to make them clearer.

Author Response

Comment #1: Authors propose a very interesting solution for SHM (Structural Health Monitoring) of structures. They suggest using mechanoluminescent materials to visualize the path of waves on the surface of a structure and use this information for SHM purposes (e.g., cracks detection or monitoring).

The paper represents a feasibility study of the concept, including a first attempt at experimental validation. This attempt failed, but the authors honestly present these results and seek to understand the reasons behind the failure of their initial trial. This paper and its content could therefore be very useful for those wishing to work in this direction and continue developing this solution.

I therefore recommend the publication of this paper, after some minor improvements.

L73-81: explanations are not clear. Please improve.

Our Reply:

Thank you for your interest and encouraging comments on our proposed novel concept and challenging research exploration.

L75-84 explains the fact that mechanoluminescence is due to localised physical processes, which can be classified according to the phenomenon involved. It would be very complex to add further explanations on this. We attempted to clarify this with revisions as highlighted with track change and added recommendation on further reading in the manuscript. The reader can refer to the following article for further explanations:

Chandra BP, Rathore AS (1995) Classification of Mechanoluminescence. Crystal Research and Technology 30:885–896

Comment #2: L91: for "stress" or "strain" monitoring?

Our Reply:

The change was made L95. Indeed, the strain is monitored to then obtain stress sensing.

Comment #3: Some comments about wear? How would such a coating (PVDF + ML particles) resist harsch environmental conditions (e.g., heavy rain, hail, salt and small stones projections...)

Our Reply:

The wear of the coating is an important question for SHM applications. The PVDF showed promising results in harsh environmental conditions when using a protective layer. This was added in the manuscript with the following reference:

Han JK, Wong V-K, Lim DBK, Subhodayam PTC, Luo P, Yao K (2023) Environmental Robustness and Resilience of Direct-Write Ultrasonic Transducers Made from P(VDF-TrFE) Piezoelectric Coating. Sensors 23:1–12

Comment #4: How would PVDF be poled on a bridge? How long would the polarization "survive", in outdoor conditions?

Our Reply:

There is portable equipment for non-contact Corona poling that could be used. This would be extra step for installation and implementation, while it is still unsure if the poling is necessary for achieving ML. The discussion was added at the end of the discussion part.

Comment #5: Fig. 4 actually shows huge discrepancies between simulated and observed results. However, authors have the merit and honesty "to show it as it is".

Our Reply:

We showed data as it is because this still confirm the order of magnitudes are corrects. With the study being at a stage for feasibility investigation, we are looking into order of magnitude to find if the pressure threshold can be potentially reached. The variations are highlighted in the manuscript L279-282.

Comment #6: 323-325: you suggest using optical fibers to pick up and transfer the dim light emitted by the coatings... All in all, this complexifies the solution too much, for practical SHM applications.

Our Reply:

The reviewer point is well understood and accepted. Of course, this would make the system more complex. It was only a suggestion for solving accessibility issues.

Comment #7: Also... if you need to mount a highly powerful PZT transducer just next to the PVDF-ML coating to excite it, you would be better off installing a standard strain sensor directly where you want to measure the stress or strain state (e.g., vibrating wire sensor, DMS gauges etc...). It hard to see what your solution would bring in that case. But again, maybe there is a niche somewhere, for your solution.

Our Reply:

We agree this solution is not optimised. We added more relevant discussions at the end of the Discussion section and at the end of the conclusions. The pressure values obtained at very short distances show that it is still not practical to achieve the outcome over large distances as we were targeting when using guided waves. Nevertheless, guided waves monitoring offers rich information compared to pure strain monitoring, which can be interrogated at schedule maintenance to follow the state of the structure.

Comment #8: So, please add, in the discussion section, a few sentences about what could realistically be done with this solution in the field of SHM. Coating a structure with PVDF-ML material seems highly unrealistic for a bridge (how would it be applied? Rapid wear issue...). Adding optical fibers to transfer a very weak light signal on a bridge is also impractical (complexity, cost, fragility of the system... competing with directly integrated optical fibers?). However, you may be able to identify some niche applications (buildings? tunnels? machines?...). That would be very interesting to know.

Our Reply:

Thank you for helping improve the manuscript. The discussions have been added at the end of the Discussion section. Monitoring hotspots as regions prone to damage may be more practical in the near future.

Comment #9: Some sentences are a bit difficult to read. I recommend, if possible, having the text improved by a native speaker. Alternatively, conduct a detailed proofreading and improve sentences here and there to make them clearer.

Our Reply:

We have done our best in improving the whole manuscript. In particular, the improvements in the abstract and the conclusions section have been highlighted with tracked changes. Thank you.

Round 2

Reviewer 1 Report

Comments and Suggestions for Authors

All is OK.

Reviewer 2 Report

Comments and Suggestions for Authors

Changing the title of the paper to much less promising and adding an explanatory paragraph at the end of the Discussion makes the paper better. It can now be published in the present form.